# Analysis of Masseter Muscle Structure in Patients with Mandibular Asymmetry Using Ultrasonic Diagnostic Equipment

**DOI:** 10.3390/bioengineering12111159

**Published:** 2025-10-26

**Authors:** Akito Umehara, Shugo Haga, Momoko Takakaze, Rika Kobayashi, Kanako Akatsuka, Misaki Yamashiro, Shiina Tatsuta, Haruhisa Nakano

**Affiliations:** Department of Orthodontics, School of Dentistry, Showa Medical University, 2-1-1 Kitasenzoku, Ota-ku, Tokyo 145-8515, Japannakanou@dent.showa-u.ac.jp (H.N.)

**Keywords:** masseter muscle, mandibular asymmetry, orthognathic surgery, ultrasonography, stiffness ratio, echo intensity, blood flow, relapse, occlusal function

## Abstract

Mandibular asymmetry often requires orthognathic surgery, and postsurgical relapse remains a concern. The masseter muscle may influence stability, but most studies have emphasized volumetric rather than qualitative changes. This prospective study evaluated 24 patients with mandibular asymmetry using ultrasonography preoperatively and at 1, 3, and 6 months after surgery. The parameters measured were thickness, stiffness ratio, echo intensity, and blood flow. The results showed significant postoperative adaptations. Masseter echo intensity increased markedly at 1 month (*p* < 0.001), peaked at 3 months (*p* = 0.042), and decreased toward baseline at 6 months (*p* < 0.001). Blood flow increased significantly from T1 to T2 (*p* < 0.001). Bite force dropped transiently at 1 month (*p* < 0.001) but recovered by 6 months (*p* < 0.001). At baseline, BMI correlated with echo intensity (r = 0.724, *p* < 0.001) and grip strength correlated with bite force (r = 0.705, *p* < 0.001). The stiffness ratio difference (contraction–rest) correlated with bite force (right: r = 0.629; left: r = 0.690). Relapse occurred in 25% of patients and correlated only with preoperative deviation and not ultrasound indices. Conclusions: Ultrasonography revealed meaningful qualitative muscle changes during recovery, though these were not strong predictors of relapse. Ultrasound remains a reliable, noninvasive tool for monitoring postoperative adaptation.

## 1. Introduction

When the mandible is asymmetric relative to the facial midline, it is known to cause not only esthetic disharmony but also the impairment of masticatory and temporomandibular joint function, significantly affecting patient quality of life (QOL). Such mandibular asymmetry becomes difficult to correct skeletally after the growth period. Therefore, in cases where orthodontic treatment alone is insufficient, combined orthodontic and orthognathic surgery is indicated to improve skeletal symmetry and occlusal relationships. However, mandibular asymmetry cases have been reported to have a high risk of postsurgical skeletal relapse, and the stability of correcting these cases has been debated for many years [1]. The causes of relapse are multifactorial, including skeletal factors (such as the amount of bone movement, fixation method, and patient growth pattern) and significant influence from surrounding soft tissues, particularly the masticatory muscles [1,2]; that is, even if the skeletal position is aligned by surgery, residual muscle tension or functional imbalance may destabilize the newly acquired jaw position. Among the masticatory muscles, the masseter is the primary elevator of the mandible and is highly likely to directly affect postoperative stability. The masseter also influences bite force, so changes in this muscle before and after surgery are clinically important [3]. Previous studies have shown that the size or volume of the masseter is related to postoperative relapse. Zhao et al. reported that, in skeletal Class III asymmetric patients, those with a larger preoperative masseter volume tended to exhibit greater relapse [2]. These findings suggest that imbalances in muscle size and force may affect skeletal stability.

In addition, side-to-side differences in the masseter have been reported in patients with mandibular asymmetry even before surgery. Goto et al. reported that, in lateral mandibular deviation cases, the masseter on the deviated side was smaller [3]. In contrast, Kwon et al. found no significant side-to-side difference in the masseter among Class III asymmetric patients [4], indicating that the findings are not entirely consistent. Furthermore, in growing patients with unilateral crossbite, the masseter on the crossbite side is thinner, but orthodontic treatment improves this and eliminates the side-to-side difference [5]. These observations indicate that the masseter exhibits plasticity in functional and morphological changes, and it is thought to readapt to skeletal changes even after orthognathic surgery.

In fact, it has been reported that, after orthognathic surgery, the masseter undergoes adaptive morphological changes over time. Using three-dimensional CT, Lee et al. showed that, in patients with mandibular asymmetry, the cross-sectional area and thickness of the masseter increased after surgery, with the side-to-side difference decreasing after 1 year and disappearing by 4 years, reaching levels comparable to those of healthy controls [6]. Zhao et al. also reported that the side-to-side difference in masseter volume significantly decreased at six months postoperatively [2]. These results indicate that, with orthognathic surgery, the masseter is gradually reconstructed over time, improving mandibular symmetry.

However, many previous studies have focused on quantitative changes, such as thickness or volume, whereas qualitative changes in the muscle have not been sufficiently examined. In recent years, advances in ultrasound imaging have enabled an in vivo qualitative evaluation of muscle properties, including stiffness ratio, muscle echo intensity, and blood flow. Sunal Aktürk et al. reported that, at three months postoperatively, the stiffness ratio and activity level of the masseter increased [7]; meanwhile, Akbulut et al. reported that masseter thickness increased along with changes in internal muscle echo intensity at six months postoperatively, suggesting tissue remodeling [8]. These reports indicate that orthognathic surgery induces not only quantitative but also qualitative changes in the masseter, which may help to explain the inconsistent findings regarding masseter thickness in patients with mandibular asymmetry.

Therefore, in this study, we longitudinally evaluated the thickness, stiffness ratio, echo intensity, and blood flow of the masseter muscle before and after orthognathic surgery in patients with mandibular asymmetry via ultrasound. By comparing values at baseline and at 1, 3, and 6 months postoperatively, we aimed to clarify how the masseter adapts qualitatively and quantitatively in order to comprehensively understand the muscle’s role in preventing postsurgical relapse, as well as its role in functional recovery.

## 2. Materials and Methods

The study included 24 patients (12 men, 12 women; age 20–58 years, mean 31.6 years) with dentofacial deformities and whose mandibular menton deviated 3.0 mm or more from the facial midline. Patients with congenital craniofacial syndromes, systemic neuromuscular disorders, or temporomandibular disorders (TMDs) that could affect muscle structure or function were excluded. All patients underwent orthognathic surgery and pre- and postoperative orthodontic treatment, and informed consent for study participation was obtained.

A Hitachi ARIETTA ultrasound device (Hitachi Ltd., Tokyo, Japan) [Figure 1] was used for evaluation. Subjects were seated with the head in a natural position, and the right and left masseter muscles were measured. The masseter thickness on each side was measured using high-resolution B-mode imaging (10 MHz linear probe) [Figure 2a,b], and muscle stiffness ratio was evaluated using strain elastography. To quantify muscle stiffness ratio, a coupler gel of known elasticity attached to the probe tip served as the reference tissue, and the stiffness ratio (ratio of strain between the masseter and the reference gel) under mild compression was calculated, with the reference gel as the denominator and the masseter as the numerator. Because the strain distribution is nonuniform depending on the region, the strain values of both the masseter and the reference gel were obtained as the mean strain within manually selected rectangular region of interest (ROIs) placed in the central, homogeneous area of each tissue and adjusted to the largest possible size within the image. Representative ROIs are shown in [Figure 2c]. Muscle echo intensity was evaluated by digitally storing the B-mode image and using ImageJ version 1.54 (National Institutes of Health, Bethesda, MD, USA; https://imagej.net/ij/, accessed on 30 August 2025) to perform a histogram analysis of grayscale intensity (mean brightness value 0–255) in the masseter region [Figure 2d]. Muscle blood flow was assessed using the color Doppler mode; the percentage of blood flow signals within the masseter region was defined as the blood flow index [Figure 2e].

Each of the above measurements was performed three times under two conditions (resting position and maximal clench), and the average of the three trials was used as the data point. Measurements were repeated at pre-surgery (T1) and 1 month (T2), 3 months (T3), and 6 months (T4) post-surgery. Measurement conditions (constant probe pressure and repeated measurements under identical conditions) were standardized according to Kobayashi et al. to confirm intra-examiner reproducibility [9]. All measurements were performed twice by the same examiner, and intra-observer reliability was confirmed using intraclass correlation coefficients (ICC = 0.86–0.91). In addition, supplementary data, including height, weight, BMI, grip strength, bite force, and occlusal contact area, were recorded at each time point. Statistical analyses were performed using JMP Pro 17.0 (SAS Institute Inc., Cary, NC, USA) with a significance level of 5%. The normality of each variable at each time point was confirmed by the Shapiro–Wilk test, and repeated measures analysis of variance (ANOVA) with time as the within-subject factor was used to detect temporal changes. To control for multiple comparisons, Bonferroni correction was applied in the post hoc tests. Pearson’s correlation coefficient was calculated to examine correlations between preoperative side-to-side differences and postoperative changes in each occlusal function parameter. In addition, we applied linear mixed-effects models (LMMs) to account for inter-individual variability, with subject-specific random intercepts and time (T1–T4) as a fixed effect, estimated using restricted maximum likelihood (REML). The LMM results were largely consistent with those of the ANOVA, supporting the robustness of our findings. Furthermore, effect sizes (partial η^2^ for ANOVA, r values for correlations) and 95% confidence intervals were reported alongside *p*-values.

## 3. Results

Of the 24 subjects, about half were in their 20 s, 25% were in their 30 s, and the remainder were in their 40 s to 50 s. Skeletal Class III was the most common classification, while Classes I and II were nearly equal in distribution. Notably, left-sided mandibular deviation was nearly five times more common than right-sided deviation [Figure 3a–c].

Regarding correlations among baseline measurements, body mass index (BMI) showed a strong positive correlation with preoperative (T1) masseter echo intensity (r = 0.724, *p* < 0.001, 95%CI [0.45, 0.87]) [Figure 4a]. There was also a strong positive correlation between grip strength (mean of left and right) and bite force (mean of left and right) (r = 0.705, *p* < 0.001, 95%CI [0.42, 0.86]) [Figure 4b]. Preoperatively (T1), the difference in muscle stiffness ratio (contraction minus rest) for both the right and left masseter showed a significant positive correlation with bite force (right: r = 0.629, *p* = 0.001, 95%CI [0.30, 0.82]; left: r = 0.690, *p* ≤ 0.001, 95%CI [0.40, 0.86]) [Figure 4c,d].

The direction and amount of mandibular deviation had a negative correlation with the side difference in occlusal contact area (r = −0.452, *p* = 0.0267, 95%CI [−072., −0.06]) [Figure 5a]. However, no significant correlation was found between the mandibular deviation (direction or amount) and side differences in masseter thickness, stiffness ratio, echo intensity, or blood flow. There was a positive correlation between side differences in the occlusal contact area and bite force (r = 0.451, *p* = 0.027, 95%CI [0.06, 0.72]) [Figure 5b]. Both the occlusal contact area difference and the bite force difference were positively correlated with the side difference in masseter thickness (occlusal contact area: r = 0.563, *p* = 0.0042, 95%CI [0.21, 0.79]; bite force: r = 0.566, *p* = 0.0039, 95%CI [0.21, 0.79]) [Figure 5c,d]. Furthermore, the side difference in the occlusal contact area was positively correlated with the side difference in the masseter blood flow (r = 0.509, *p* = 0.011, 95%CI [0.13, 0.76]) [Figure 5e].

The side difference values (occlusal contact area, bite force, masseter thickness, blood flow index) were calculated as right–left, as shown in Figure 5. A positive value indicates greater measurement on the right, while a negative value indicates greater measurement on the left.

From pre-surgery to post-surgery, masseter echo intensity increased significantly from T1 to T2 and increased slightly further at T3, but it had significantly decreased back to nearly the preoperative level by T4 (T1→T2: *p* < 0.001, ηp^2^ = 0.97, 95% CI [0.94, 0.99]; T2→T3: *p* = 0.042, ηp^2^ = 0.87, 95% CI [0.73, 0.94]; T3→T4: *p* < 0.001, ηp^2^ = 0.50, 95% CI [0.18, 0.75]) [Figure 6a]. Masseter blood flow also significantly increased from T1 to T2 (*p* < 0.001, ηp^2^ = 0.96, 95% CI [0.92, 0.98]) [Figure 6b]. Bite force showed a transient decrease at 1 month post-operation (T2) but recovered over time, approaching the preoperative value by 6 months (T4) (T1→T2: *p* < 0.001, ηp^2^ = 0.95, 95% CI [0.89, 0.98]; T2→T3: *p* < 0.001, ηp^2^ = 0.17, 95% CI [0.00, 0.49]; T3→T4: *p* < 0.001, ηp^2^ = 0.84, 95% CI [0.66, 0.93]) [Figure 6c].

There were no significant changes in the side differences in masseter thickness, stiffness ratio, echo intensity, or blood flow from T1 to T4.

Relapse (Because this study focused on observing how masseter asymmetry influences mandibular asymmetry in patients with lateral deviation, relapse was defined solely as lateral mandibular displacement. Considering that maxillary relapse or anteroposterior mandibular changes would have complicated the analysis, these components were not included in this study.) was evaluated at 6 months post-operation (T4) and was observed in 6 of the 24 patients. The amount of mandibular lateral relapse showed a positive correlation with the preoperative deviation amount (r = 0.562, *p* = 0.0042, 95%CI [0.21, 0.79]), but no correlation was found between relapse and the masseter’s thickness, stiffness ratio, echo intensity, or blood flow [Figure 7].

## 4. Discussion

**BMI and masseter echo intensity:** The strong positive correlation between BMI and masseter echo intensity suggests that patients with a higher BMI may have increased fat infiltration or fibrosis within the masseter. Previous studies have shown that echo intensity correlates with the proportion of fat or fibrous tissue in muscle [10,11]. Obesity, therefore, may lead not only to increased body weight but also to histological changes in the masticatory muscles, potentially resulting in muscle dysfunction and affecting post-orthognathic surgery stability. However, these findings indicate that BMI may have acted as a confounding factor in our study. Due to the limited sample size, statistical adjustments such as ANCOVA or regression analysis were not feasible. Future studies with larger cohorts should incorporate BMI as a covariate to clarify its influence.**Muscle function and overall muscle strength:** The observed positive correlation between grip strength and bite force, as well as the relationship between the stiffness ratio difference (contraction minus rest) and bite force, indicates synergy between the overall muscle strength and localized masticatory function. Previous studies have used grip strength as an indicator of general muscle strength, finding it to be related to masticatory function and nutritional status [12]. Recent research in children also found a significant correlation between grip strength and tongue pressure [13], supporting a link between overall muscle strength and oral function. Regarding muscle stiffness ratio, when considering factors like aging and contracture, there is little consistent evidence on what stiffness ratio reflects. Our results suggest that, when the masseter is relatively soft at rest, a greater increase in stiffness ratio during contraction is associated with higher force generation.**Mandibular asymmetry and occlusion–masseter relationships:** Previous studies have not reached a consensus on the relationship between mandibular asymmetry and masticatory muscle activity or conditions. For example, some have reported that, in mandibular asymmetry cases, the masseter on the deviated side is thinner and less active [3]. Matsuda et al. observed that, during clenching, the temporalis on the deviated side and the masseter on the non-deviated side tended to be dominant [14], and Beltrami et al. reported that, in crossbite patients, the masseter on the crossbite (deviated) side was thinner than on the non-crossbite side [5]. Other studies have found conflicting results, including that the masseter is thicker on the deviated side, thicker on the non-deviated side, or shows no side difference, leading to confusion. In the present study, masseter thickness was more strongly associated with the occlusal contact area than with the direction of mandibular deviation. Specifically, patients who had a greater occlusal contact area on one side tended to have increased masseter thickness on that side, regardless of the deviation direction; that is, patients who habitually chew on one side (the side with more occlusal contact) tend to develop a stronger bite force and masseter hypertrophy on that side, where it is known that habitual unilateral chewing leads to masseter hypertrophy on the favored side [15]. Nakano et al. found that shifting the mandible laterally in a rat model changed the shape and trabecular bone of the mandible and condyle prior to changes in the masseter muscle [16]. These findings suggest that daily masticatory load may have a greater effect on masseter thickness than the mandible’s skeletal deviation direction. However, the causal relationship between mandibular asymmetry and occlusal deviation remains unclear: it is not certain whether a skeletal shift causes occlusal deviation or whether an asymmetric occlusion leads to or worsens mandibular asymmetry. This issue was not directly tested in the present study and remains a subject for future research.**Postoperative changes in the masseter:** The observed increase in echo intensity at 1 month post-operation (T2) is likely due to inflammation, edema, or fibrosis, and its subsequent recovery over time reflects tissue repair processes. A similar pattern of echogenic change has been reported in prior studies on muscle healing [17]. The early increase in blood flow is consistent with the circulatory response and healing process after surgical trauma, as reported by Salmi et al. [18]. Bite force exhibited an early postoperative decline followed by recovery by 6 months, likely reflecting a temporary reduction in muscle function due to surgical trauma and the avoidance of chewing due to pain, followed by muscle remodeling and functional recovery. Previous reports have similarly shown a transient decrease in bite force after surgery, with gradual improvement over time [19], which is consistent with our findings. Bite force recovery may be influenced by postoperative diet and rehabilitation and is an important indicator of long-term stability [20].**Masseter-side difference progression:** In this study, no significant changes were observed in the side differences in masseter thickness, stiffness ratio, echo intensity, or blood flow from before surgery to 6 months post-operation. This is consistent with previous reports suggesting that muscle plasticity and functional adaptation may continue over a period of 1 year to 4 years after surgery [6]. Our study is ongoing, and future follow-up will include data beyond 6 months to comprehensively evaluate these long-term changes.**Relation to relapse amount:** The relapse rate in this study was 25% (6 of 24 patients), which is similar to the proportion reported by Yang et al. [21]. The number of relapse cases at 6 months was correlated with the preoperative mandibular deviation amount; that is, cases with larger initial deviation (requiring larger surgical movements) tended to have a higher risk of relapse, which is consistent with the findings from Lai et al. [22].

### Limitations

This study included a relatively small sample (n = 24, with only six relapse cases), which limits statistical power; small-to-moderate effects may therefore have gone undetected. The low number of relapse events also precluded robust multivariable modeling to identify independent predictors. Accordingly, the findings of this study should be regarded as preliminary evidence that provides a foundation for future large-scale investigations.Because the primary aim of this study was to evaluate longitudinal changes in patients with mandibular asymmetry, a healthy symmetric control group was not included. This absence represents a limitation, as it prevents the establishment of normative baseline values and may restrict the clinical interpretation of our findings.To avoid overcomplicating the analysis, relapse in this study was defined solely as lateral mandibular displacement. This restricted definition meant that vertical and anteroposterior relapse were not assessed, which represents one of the limitations of this study.Although ultrasonography is a safe, real-time, and noninvasive method for evaluating the masseter, it has inherent limitations compared with MRI or CT. Previous studies have demonstrated that ultrasound can provide results comparable to MRI and CT under standardized conditions [23,24,25], but inter-operator variability and probe positioning remain challenges. Recent innovations, such as AI-assisted ultrasound analysis and 3D/AR elastography, have been reported to improve reproducibility and enhance the accuracy of anatomical localization, even for less experienced operators. Yan et al. summarized advances in AI-assisted ultrasound diagnosis [26], Costa et al. presented an AR-based ultrasound guidance system for breast biopsy [27], and Ng et al. developed a portable mixed-reality 3D ultrasound tracking and reconstruction system [28]. In line with these developments, our findings may further broaden the clinical utility of ultrasound in postoperative evaluation and relapse prediction when combined with such cutting-edge technologies.Postoperative rehabilitation and physiotherapy may also influence the recovery and adaptation of the masticatory muscles. Previous studies have suggested that targeted rehabilitation programs can facilitate functional recovery after orthognathic surgery [29,30]. However, because standardized rehabilitation data were not collected in our study, this represents another limitation.

## 5. Conclusions

This study did not find a clear correlation between the number of postsurgical relapse cases and preoperative side-to-side differences in quantitative or qualitative masseter parameters. As previous studies have indicated, relapse is multifactorial [1,2], making it difficult to interpret the relationship based on muscle information alone. On the other hand, this study demonstrated that ultrasound evaluation of the masseter is effective not only for quantitative but also for qualitative assessment, as evidenced by the significant correlation between masseter stiffness ratio and bite force, as well as the consistent postoperative changes in echo intensity and blood flow. With further long-term follow-up, if the timing and sequence of improvement in side-to-side differences in masseter thickness, stiffness ratio, echo intensity, and blood flow become clearer in relation to relapse, ultrasound assessment of the masseter could serve as a useful indicator for predicting postsurgical relapse risk. Furthermore, by utilizing these ultrasonographic parameters, a data-driven, machine learning-based classifier could potentially be developed to predict postsurgical relapse.

## Figures and Tables

**Figure 1 bioengineering-12-01159-f001:**
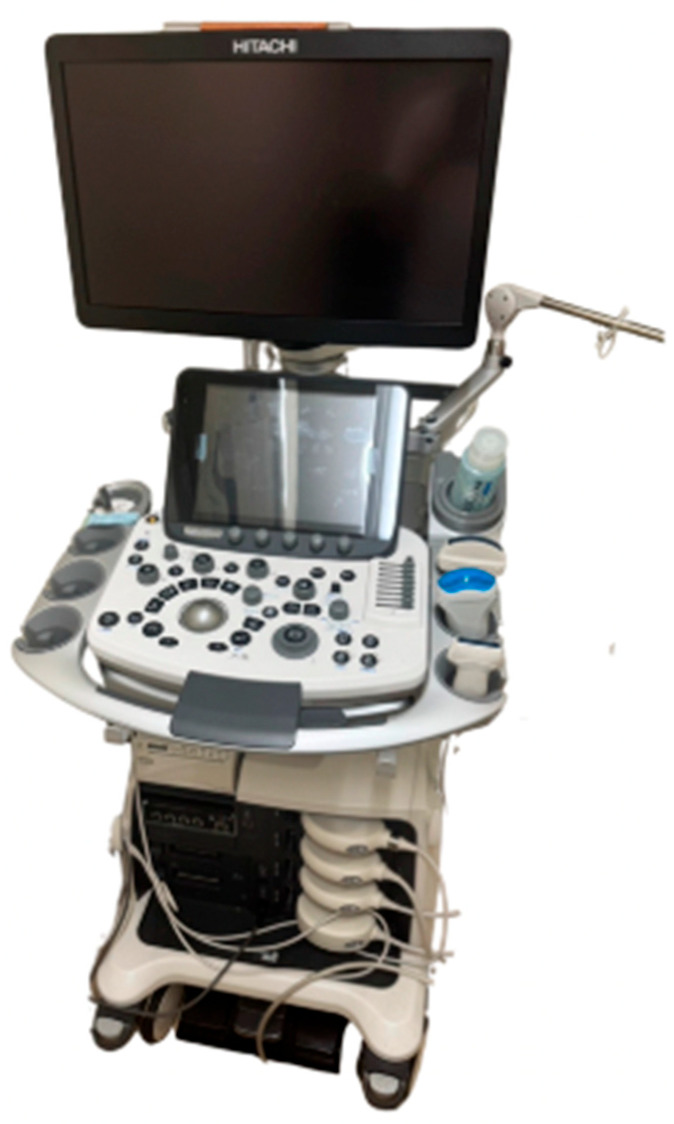
Ultrasound diagnostic system (Hitachi ARIETTA).

**Figure 2 bioengineering-12-01159-f002:**
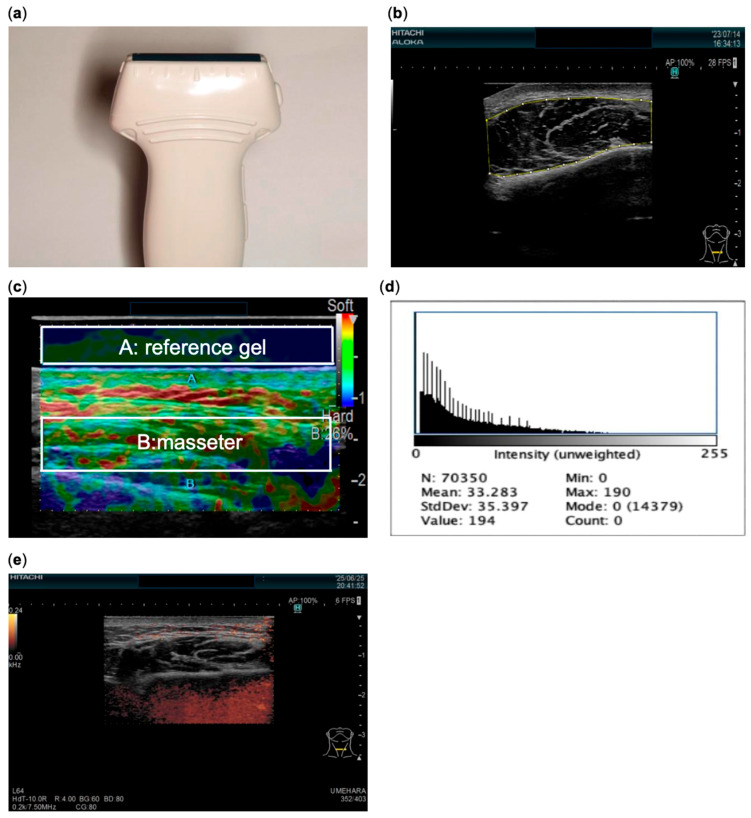
(**a**) A10 MHz linear probe. (**b**) B-mode image. (**c**) Stiffness ratio calculated using a coupling gel pad as the reference tissue, with the reference gel (A) as the denominator and the masseter (B) as the numerator, expressed as the ratio of strain between the masseter and the reference gel. (**d**) Histogram analysis of masseter muscle echo intensity (grayscale intensity, MGV) using ImageJ. (**e**) Blood flow index measured with color Doppler mode.

**Figure 3 bioengineering-12-01159-f003:**
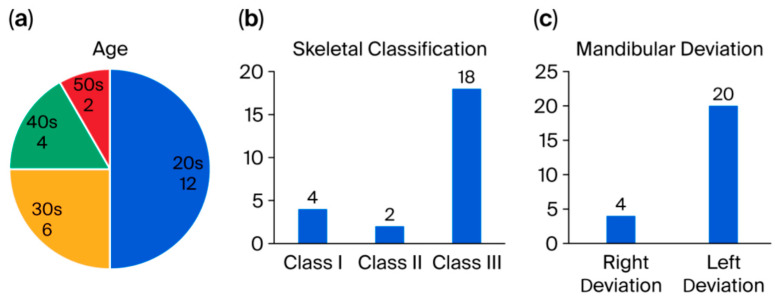
(**a**) Age distribution of the 24 participants. (**b**) Skeletal classification of the 24 participants (Class I: ANB 0–4.0°; Class II: ANB ≥ 4.0°; Class III: ANB ≤ 0°). (**c**) Mandibular deviation direction of the 24 participants.

**Figure 4 bioengineering-12-01159-f004:**
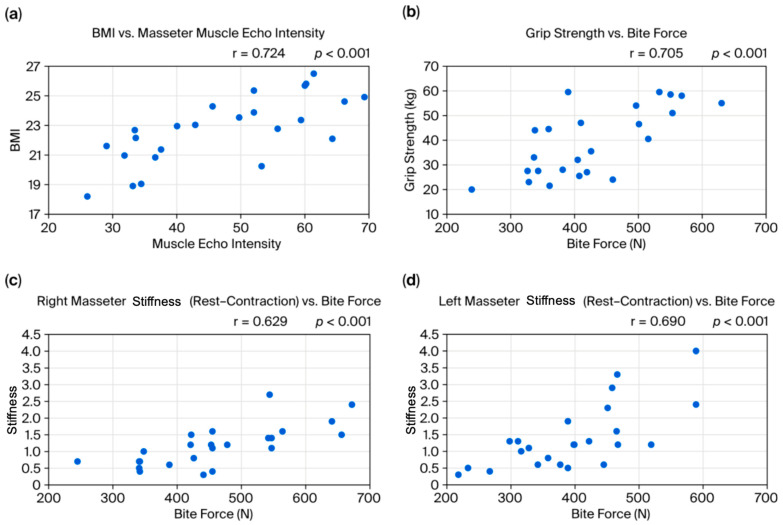
(**a**) Positive correlation between BMI and masseter muscle echo intensity. (**b**) Positive correlation between grip strength and bite force. (**c**) Positive correlation between right masseter stiffness ratio (rest–contraction) and bite force. (**d**) Positive correlation between left masseter stiffness ratio (rest–contraction) and bite force.

**Figure 5 bioengineering-12-01159-f005:**
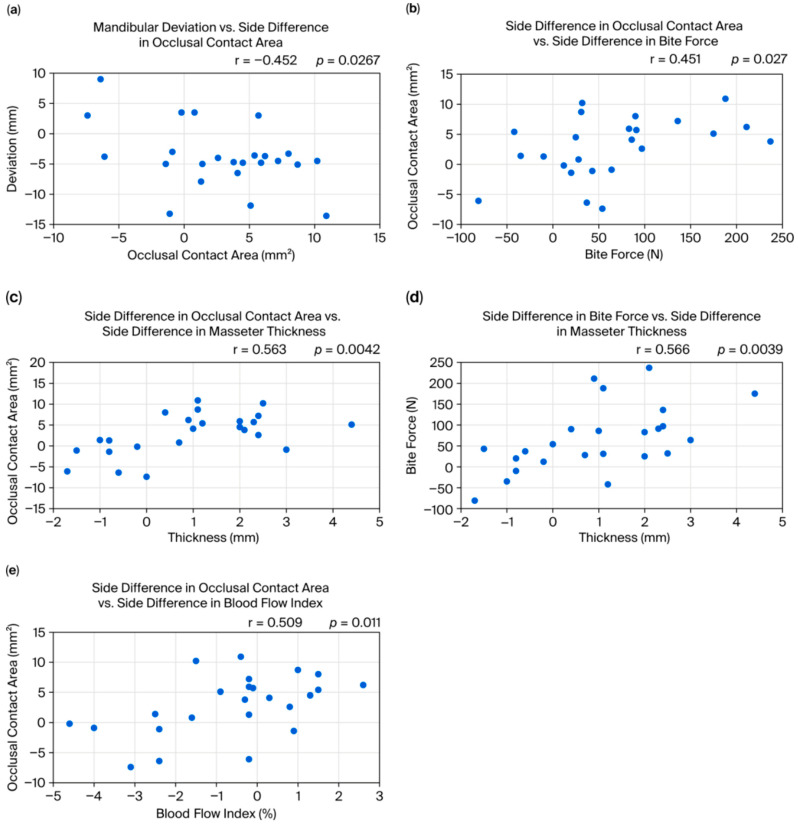
(**a**) Negative correlation between mandibular deviation and side difference in occlusal contact area. (**b**) Positive correlation between side difference in occlusal contact area and side difference in bite force. (**c**) Positive correlation between side difference in occlusal contact area and side difference in masseter thickness. (**d**) Positive correlation between side difference in bite force and side difference in masseter thickness. (**e**) Positive correlation between side difference in occlusal contact area and side difference in blood flow index.

**Figure 6 bioengineering-12-01159-f006:**
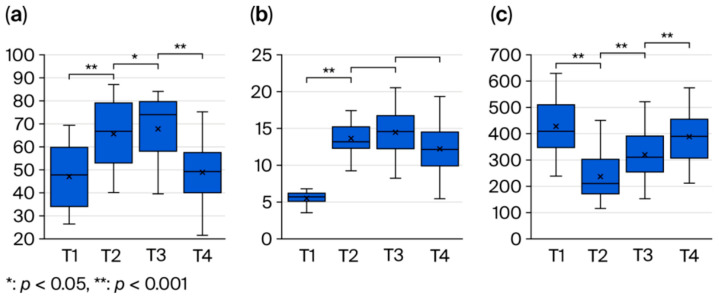
(**a**) Longitudinal changes in masseter muscle echo intensity (T1–T4). (**b**) Longitudinal changes in masseter blood flow index (T1–T4). (**c**) Longitudinal changes in bite force (T1–T4).

**Figure 7 bioengineering-12-01159-f007:**
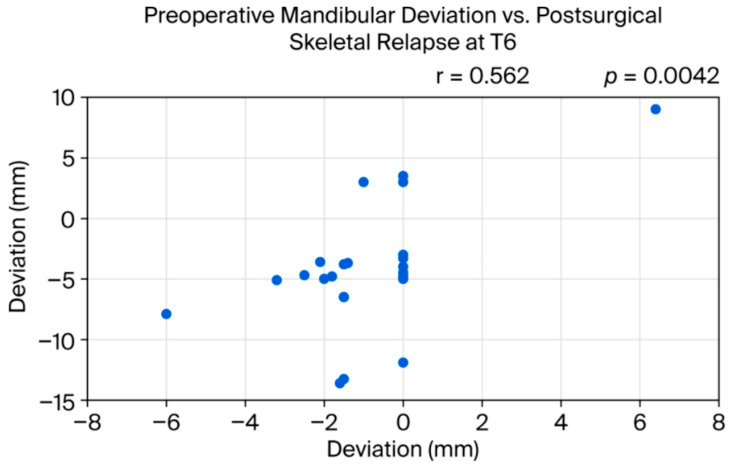
Positive correlation between preoperative mandibular deviation and postsurgical skeletal relapse at T6.

## Data Availability

The original contributions presented in the study are included in the article, further inquiries can be directed to the corresponding author.

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
