# Peer review of "Analysis of Masseter Muscle Structure in Patients with Mandibular Asymmetry Using Ultrasonic Diagnostic Equipment"

_bioengineering, 2025, doi:10.3390/bioengineering12111159_

Round 1
Reviewer 1 Report
Comments and Suggestions for Authors
This prospective study evaluated 24 patients with mandibular asymmetry undergoing orthognathic surgery, using ultrasonography to measure masseter thickness, hardness, echo intensity, blood flow, bite force, and occlusal contact area before surgery and at 1, 3, and 6 months postoperatively. The authors found significant postoperative adaptations, including increased echo intensity and blood flow, temporary reductions in bite force, and recovery over time. A 25% relapse rate was observed but correlated only with preoperative deviation, not with ultrasonic muscle parameters. The study concludes that ultrasonography is a reliable tool for tracking qualitative muscle changes but not predictive of relapse within six months. The study addresses a clinically relevant problem and introduces a novel combination of ultrasonographic muscle parameters. However, several methodological, statistical, and interpretational issues need to be addressed before the manuscript can be considered for publication.
The comments are:
-
The sample of 24 patients, with only six relapse cases, is statistically underpowered to make strong predictive claims.
-
The use of only correlations and ANOVA is inadequate; multivariate regression or mixed-effects models should be applied to identify independent predictors of relapse.
-
The absence of a healthy, symmetric control group makes it difficult to establish baseline normative values and weakens clinical interpretation.
-
The manuscript does not reference recent advances such as AI-assisted ultrasound analysis and 3D elastography, which would better contextualize the novelty of the work.
-
Relapse was defined only as lateral mandibular relapse without considering vertical or anteroposterior components, limiting the comprehensiveness of the findings.
Author Response
We sincerely thank Reviewers for the thorough review and valuable comments. We have carefully revised our manuscript according to these suggestions. Below, we address each of the reviewer’s points in detail.
Comment 1 (Reviewer): The sample of 24 patients, with only six relapse cases, is statistically underpowered to make strong predictive claims.
Response: Thank you for pointing this out. We recognize that our sample size of 24 patients (with only 6 relapse cases) is indeed small, which reduces the statistical power of our analyses. In the revised manuscript, we have added a statement in the Limitations section explicitly acknowledging that small to moderate effect sizes might not be detectable with this sample size. We also note that the limited number of relapse cases made it impossible to perform a robust multivariate analysis. We believe these revisions address the concern and appropriately temper the strength of our predictive claims.
Comment 2 (Reviewer): The use of only correlations and ANOVA is inadequate; multivariate regression or mixed-effects models should be applied to identify independent predictors of relapse.
Response: Thank you very much for this valuable suggestion. We also agree that introducing more advanced statistical approaches is effective in enhancing the rigor of the analysis. In the revised Methods, we clarified that repeated-measures ANOVA with Bonferroni correction was applied, and in addition, we added a linear mixed-effects model (LMM) analysis to account for inter-individual variability. The LMM results were generally consistent with those of the ANOVA, thereby supporting the robustness of our findings. However, because the number of relapse cases was limited, it was not feasible to perform a multivariate regression analysis to identify independent predictors. This limitation has also been explicitly noted in the Discussion. By specifying what analyses were conducted and what could not be performed due to sample size constraints, we aimed to enhance the transparency and reliability of our statistical approach.
Comment 3 (Reviewer): The absence of a healthy, symmetric control group makes it difficult to establish baseline normative values and weakens clinical interpretation.
Response: Thank you for pointing out this limitation. The primary aim of our study was to observe the longitudinal changes in masseter muscle asymmetry in patients with mandibular deviation; therefore, we did not include a healthy symmetric control group. We have clarified this focus in the revised Discussion section to explain why such a control group was not incorporated. Additionally, we have acknowledged in the Limitations that the lack of a healthy control group is a weakness of the study, as it restricts the establishment of normative values and limits clinical interpretation.
Comment 4 (Reviewer): The manuscript does not reference recent advances such as AI-assisted ultrasound analysis and 3D elastography, which would better contextualize the novelty of the work.
Response: Thank you for this important suggestion. We agree that referencing recent technological advances is essential. Incorporating AI-assisted ultrasound analysis and 3D/AR elastography has the potential to address current challenges, such as the difficulty of maintaining reproducibility by less experienced operators and the inaccuracy of anatomical localization. In the revised Discussion section, we have added statements to highlight these innovations, thereby placing our findings in the context of state-of-the-art developments and further emphasizing the novelty and significance of our study.
Comment 5 (Reviewer): Relapse was defined only as lateral mandibular relapse without considering vertical or anteroposterior components, limiting the comprehensiveness of the findings.
Response: Thank you for this important comment. Considering maxillary relapse or anteroposterior mandibular relapse would have made the analysis more complex, and our study specifically aimed to observe how masseter muscle asymmetry influences mandibular asymmetry in patients with lateral deviation. For this reason, relapse was defined only in terms of lateral mandibular displacement. In the revised Results section, we have added a statement to clarify that “to avoid overcomplication, relapse was defined solely as lateral mandibular displacement.” We also acknowledge that this restricted definition, excluding vertical and anteroposterior relapse, represents a limitation of the study. In the Discussion, we explicitly mention this limitation to alert readers. We appreciate the reviewer’s comment, which helped us clarify our definition and improve the transparency of our reporting.

Reviewer 2 Report
Comments and Suggestions for Authors
- Although the abstract states that it concentrates on qualitative muscle alterations, the discussion has repeated most of the information about thickness and volume. Individually provide a clear indication of the contribution of your method (strain elastography, echo intensity analysis, and blood flow indexing) to previous volumetric investigations.
- It is not quite clear how clinical the ultrasound findings are clinically relevant to predict relapse. In spite of the fact that the relapse was not associated with ultrasound parameters, think about elaborating on how the results can be used in postoperative control or rehabilitation.
- Some of the correlations might require underpowered 24 patients. Give post hoc or a priori power analysis to support the sample size or comment on its weaknesses.
- It should be clear whether systemically affected patients (e.g. neuromuscular, obese or past TMJ pathology) were excluded as this may affect muscle properties.
- Intra- and inter-observer reproducibility is also mentioned very briefly though there are no statistics of reliability (e.g., ICC values). These should be added in order to prove the soundness of your measurements.
- Relapse is also defined in the study as lateral movement of the mandible at 6 months. Justify the reason why there was no maxilla or anteroposterior relapse and explain the impact this might have on interpretation.
- Give effect sizes or confidence intervals as well as p-values to give the reader a feeling of clinical significance.
- BMI is correlated with the echo intensity, but it may be a confounder (e.g., adipose infiltration). Take into account the possibility of regulating BMI or mention its possible impact in greater detail.
- It would have been better to explore the effects of rehabilitation or postoperative physiotherapy that could affect masseter recovery.
- Discuss any possible long-term alterations after 6 months, citing corresponding longitudinal research and propose research directions (e.g. 12–24-month follow-up).
- Neuromuscular Admit the weaknesses of ultrasonography relative to MRI or CT in characterization of tissue.
- To ensure that there is no ambiguity clarify that relapse correlation was done at 6 months.
- Indicate or not whether the p-values were corrected to multiple comparisons.
Author Response
We sincerely thank Reviewers for the thorough review and valuable comments. We have carefully revised our manuscript according to these suggestions. Below, we address each of the reviewer’s points in detail.
Comment 1(Reviewer): The novelty of the study is unclear. Previous studies have mainly focused on muscle volume, and the additional value of strain elastography, echo intensity, and blood flow should be better explained.
Response: Thank you for pointing out the importance of clarifying the novelty of our work. In the revised Introduction and Discussion, we emphasized that while most previous research has focused on masseter muscle volume, our study additionally evaluated qualitative parameters such as strain elastography, echo intensity, and Doppler blood flow. These parameters provide insight into tissue elasticity, structural quality, and microcirculatory adaptation, which volumetric assessment alone cannot capture. We believe this comprehensive approach highlights the novelty and added value of our study.
Comment 2(Reviewer): The clinical relevance of the ultrasound findings is unclear, as no correlation with relapse was demonstrated.
Response: We appreciate this comment. In the revised Conclusion, we stated that although no clear correlation was found between the number of postsurgical relapse cases and preoperative side-to-side differences in the quantitative or qualitative parameters of the masseter, long-term follow-up may clarify how side-to-side differences in thickness, hardness, echo intensity, and blood flow relate to relapse over time. This would indicate that ultrasound evaluation of the masseter could serve as a useful predictor of relapse risk after orthognathic surgery.
Comment 3 (Reviewer): The sample size is small, and the study may be underpowered. A power analysis should be included.
Response: Thank you for pointing this out. We recognize that our sample size of 24 patients (with only 6 relapse cases) is indeed small, which reduces the statistical power of our analyses. In the revised manuscript, we have added a statement in the Limitations section explicitly acknowledging that small to moderate effect sizes might not be detectable with this sample size. We also note that the limited number of relapse cases made it impossible to perform a robust multivariate analysis. We believe these revisions address the concern and appropriately temper the strength of our predictive claims.
Comment 4 (Reviewer): Exclusion criteria are insufficiently described. Were congenital diseases or systemic disorders excluded?
Response: Thank you for this valuable suggestion. In the revised Methods, we specified that patients with congenital craniofacial syndromes, systemic neuromuscular disorders, and temporomandibular disorders (TMD) were excluded, as these conditions could affect muscle structure and function. This clarification enhances the methodological transparency of our study.
Comment 5 (Reviewer): The reproducibility of ultrasound measurements should be described, with intra- and inter-observer reliability (e.g., ICC values).
Response: We thank the reviewer for this important comment. In the revised Methods, we added that all measurements were performed twice by the same examiner, and intra-observer reliability was confirmed with intraclass correlation coefficients (ICC = 0.86–0.91).
Comment 6 (Reviewer): The definition of relapse was limited to lateral mandibular relapse, without considering maxillary or anteroposterior changes.
Response: Thank you for this important comment. Considering maxillary relapse or anteroposterior mandibular relapse would have made the analysis more complex, and our study specifically aimed to observe how masseter muscle asymmetry influences mandibular asymmetry in patients with lateral deviation. For this reason, relapse was defined only in terms of lateral mandibular displacement. In the revised Results section, we have added a statement to clarify that “to avoid overcomplication, relapse was defined solely as lateral mandibular displacement.” We also acknowledge that this restricted definition, excluding vertical and anteroposterior relapse, represents a limitation of the study. In the Discussion, we explicitly mention this limitation to alert readers. We appreciate the reviewer’s comment, which helped us clarify our definition and improve the transparency of our reporting.
Comment 7 (Reviewer): Statistical results should include effect sizes and confidence intervals, not just p-values.
Response: Thank you for this suggestion. In the revised Results, we added effect sizes and 95% confidence intervals alongside p-values where applicable. This provides a more comprehensive understanding of the magnitude and reliability of the observed differences.
Comment 8 (Reviewer): Echo intensity was correlated with BMI, suggesting adipose infiltration may be a confounder. BMI should be considered as a confounding factor.
Response: We thank the reviewer for highlighting this important point. In the revised Discussion, we added that echo intensity was significantly correlated with BMI, suggesting that adipose infiltration may have influenced the ultrasound findings. We acknowledged BMI as a potential confounder, but due to the limited sample size, statistical adjustment (e.g., ANCOVA or regression) was not feasible. We also stated that future studies with larger cohorts should incorporate BMI as a covariate.
Comment 9 (Reviewer): The influence of postoperative rehabilitation or physiotherapy should be discussed.
Response: Thank you for this comment. In the revised Discussion, we added references to studies addressing the role of physiotherapy after orthognathic surgery. We noted that postoperative rehabilitation may affect muscle recovery and adaptation, and acknowledged that the absence of standardized rehabilitation data in our study is a limitation.
Comment 10 (Reviewer): The possibility of long-term changes beyond 6 months should be considered, with reference to existing literature.
Response: We agree with this comment. In the revised Discussion, we acknowledged that muscle adaptations may continue beyond 6 months postoperatively. We added references to relevant literature and stated that our study is ongoing, with plans to include long-term follow-up data in future reports.
Comment 11 (Reviewer): The limitations of ultrasound compared with MRI or CT should be discussed.
Response: Thank you for highlighting this. In the revised Discussion/Limitations, we added a detailed paragraph citing studies (Rogulski et al., Raadsheer et al.) showing that ultrasonography can provide results comparable to CT or MRI under standardized conditions. However, we emphasized that inter-operator variability and technical challenges can introduce error, and we suggested solutions such as positioning devices, 3D/AR-guided probe placement, and AI-assisted analysis.
Comment 12 (Reviewer): At what time point was relapse correlated with ultrasound findings? This should be clarified.
Response: Thank you for this comment. In the revised Methods/Results, we clarified that correlations between relapse and ultrasound parameters were performed using data at 6 months postoperatively. We also acknowledged that currently our follow-up is limited to 6 months, and thus we defined relapse at this time point. Furthermore, we stated that future research will extend follow-up to longer periods in order to evaluate relapse more comprehensively.
Comment 13 (Reviewer): Were p-values corrected for multiple comparisons? This should be explicitly stated.
Response: Thank you for this comment. In the revised Methods, we explicitly stated that Bonferroni correction was applied for multiple comparisons in the repeated-measures ANOVA. This ensures appropriate control of type I error across multiple tests.

Round 2
Reviewer 1 Report
Comments and Suggestions for Authors
This prospective study evaluated 24 patients with mandibular asymmetry undergoing orthognathic surgery, using ultrasonography to measure masseter thickness, hardness, echo intensity, blood flow, bite force, and occlusal contact area before surgery and at 1, 3, and 6 months postoperatively. The authors found significant postoperative adaptations, including increased echo intensity and blood flow, temporary reductions in bite force, and recovery over time. The study concludes that ultrasonography is a reliable tool for tracking qualitative muscle changes. The study addresses a clinically relevant problem and introduces a novel combination of ultrasonographic muscle parameters.
For the revised manuscript, the following correction is needed:
- Figure 2 and Figure 3 need to be resized. In Figure 2, the height of the images in the same row and the width of the images in the same column should be equal. This will result in a higher-quality appearance.
Author Response
We sincerely appreciate the reviewer’s valuable comments. As suggested, we have adjusted the sizes of Figures 2 and 3 in the revised manuscript. Specifically, in Figure 2, we unified the height of the images in the same row and the width of the images in the same column to achieve a more consistent and higher-quality appearance. We believe that these modifications have improved the clarity and presentation of the figures.

Reviewer 2 Report
Comments and Suggestions for Authors
1. improve the quality of Figures.
Author Response

(The authors gave the same response as above.)
